# Comprehensive Treatment of a Skeletal Class III Growing Patient with Alveolar Bone Hypertrophy in the Anterior Mandible Associated with Becker’s Nevus Syndrome: A Case Report

**DOI:** 10.3390/children8020072

**Published:** 2021-01-21

**Authors:** Won-Kyeong Baik, Sung-Hwan Choi

**Affiliations:** 1Department of Orthodontics, Institute of Craniofacial Deformity, Yonsei University College of Dentistry, Seoul 03722, Korea; wkbaik1@naver.com; 2BK21 FOUR Project, Yonsei University College of Dentistry, Seoul 03722, Korea

**Keywords:** Becker’s nevus, Becker’s nevus syndrome, alveolar bone hypertrophy, intrusion

## Abstract

Becker’s nevus is a relatively common skin lesion with brown hyperpigmentation and hypertrichosis. It may be expressed simultaneously with other cutaneous, muscular, or skeletal defects, such as hypoplasia of the breast or scoliosis, for which the term “Becker’s nevus syndrome” has been proposed. An 8-year-old boy presented with a Becker’s nevus on the right cheek, chin, and upper neck with alveolar bone hypertrophy in the anterior mandible, which led to an anterior crossbite and severe canting of the mandibular anterior teeth. Through orthopedic treatment using a face mask, the anteroposterior jaw relationship was improved. In phase 2 treatment, we corrected the canting of the mandibular anterior teeth with a segmental intrusion of the mandibular anterior right teeth into the hypertrophic alveolar bone area. The facial profile was improved along with acceptable occlusion, and the treatment result was stable at 1 year after the end of the treatment, without any periodontal attachment loss, root resorption of mandibular anterior teeth, or increase in the size of the hypertrophic region. This case report describes the successful comprehensive treatment of a growing patient with skeletal Class III malocclusion and multiple characteristics of Becker’s nevus syndrome.

## 1. Introduction

Becker’s nevus is a hyperpigmented skin lesion with hypertrichosis that was first reported by Becker in 1949 [1]. It is characterized by a patch of light or dark brown hyperpigmentation with a sharp, but irregular, outline and tends to appear in a checkerboard pattern [2,3]. Becker’s nevus is a relatively common cutaneous hamartoma with a reported incidence of 0.5% to 4% [4,5,6,7], but it is often overlooked. Since it is an androgen-dependent skin anomaly [8], it is more prevalent in men and mainly appears in the first and second decades of life, becoming more prominent after puberty [9]. Becker’s nevus has a very low risk of malignant transformation, and it does not require treatment except for cosmetic needs, with the therapeutic options including waxing, electrolysis, and laser treatment [10].

Several studies have reported the relationship between Becker’s nevus and other developmental anomalies [11,12]. In 1997, Happle and Koopman suggested a new term of “Becker’s nevus syndrome” for the simultaneous presence of a Becker’s nevus and other cutaneous, muscular, or skeletal defects, such as hypoplasia of the breast or scoliosis [13,14]. Most of these anomalies tend to coincide with the nevus sites and are mostly ipsilateral [15]. In addition, several studies have presented cases wherein maxillofacial anomalies have occurred along with the Becker’s nevus including facial asymmetry, unilateral maxillary gingival and alveolar bone hypertrophy, and tooth anomalies [16,17,18]. However, to date, a few cases of Becker’s nevus causing malocclusion by affecting the mandible have been reported in young patients.

The present case report describes the successful orthopedic and orthodontic treatment of a young boy who exhibited a skeletal Class III anteroposterior jaw relationship and had Becker’s nevus with alveolar bone hypertrophy in the anterior right mandible. Written informed consent was obtained from the patient and the parents for the publication of this case report and any accompanying images.

## 2. Case Presentation

### 2.1. Diagnosis and Etiology

An 8-year-old boy visited the orthodontic department at Yonsei University Dental Hospital in Seoul, Korea, with a chief complaint of anterior crossbite and crowding. He had undergone surgery for the removal of a supernumerary tooth in the maxillary anterior area 2 months prior to his visit. Pre-treatment facial photographs showed a midfacial deficiency and a concave profile. There was no remarkable mandibular deviation; however, he showed significant lip canting while smiling with the higher level at the right corner of the mouth. In addition, a well-defined asymptomatic hyperpigmented patch with slight hypertrichosis was present on the right cheek, chin, and upper neck, which was diagnosed as Becker’s nevus (Figure 1 and Figure 2).

Intraoral examination showed anterior crossbite and severe deep bite with a −2.0 mm overjet and 7.0 mm overbite with no functional shift. The patient exhibited severe crowding on maxillary dentition along with a maxillary transverse deficiency. The mean difference in the width of the maxillary and mandibular first molars at the age of 8 years has been reported to be 6.4 mm, whereas in our patient, it was only 4 mm [19]. He showed a 1 mm deviation of the mandibular dental midline toward the left side. Hypertrophy of the mandibular right anterior alveolar bone was clear, which had led to the canting of the mandibular anterior teeth.

Lateral cephalometric analysis showed an angle of the lines connecting the sella, nasion, and point A (SNA) of 75.4°, an angle of the lines connecting the sella, nasion, and point B (SNB) of 75.7°, and angle of the lines connecting point A, nasion, and point B (ANB) of −0.3° (Table 1 and Figure 3). Both maxillary and mandibular incisors were lingually inclined, and the upper lip was retruded with respect to the Ricketts’ esthetic line. In posteroanterior cephalometric analysis, there were no noticeable asymmetric features. A panoramic radiograph showed that a supernumerary tooth was impacted in the right mandibular premolar area. There were no pathologic osseous findings in the gingival hypertrophic area of the anterior mandible.

### 2.2. Treatment Objectives

On the basis of the clinical and radiographic findings, this patient was diagnosed with skeletal Class III malocclusion with anterior crossbite and crowding. The following treatment objectives were planned: (1) the periodic follow-up of the hypertrophic alveolar bone area, (2) an improvement in the skeletal Class III anteroposterior jaw relationship, (3) the relief of the anterior crossbite, (4) making space on the maxillary dentition for guiding eruption, (5) axis improvement of the retroclined maxillary and mandibular anterior teeth, (6) the correction of the dental midline, and (7) the extraction of the impacted supernumerary tooth.

### 2.3. Treatment Alternatives

Based on the treatment objectives, the following treatment alternatives were considered: (1) orthopedic treatment with maxillary protraction using a face mask, or (2) orthognathic surgery after the completion of skeletal growth.

The main cause of the patient’s skeletal Class III relationship was the undergrowth of the maxilla in contrast to the normal mandibular growth. Furthermore, the anterior crossbite was mainly due to the severe lingual inclination of the maxillary anterior teeth, and therefore, the prognosis with orthopedic treatment was considered to be favorable. Considering the patient’s young age as well, Option 1 was chosen with the consent of the patient and the parents.

### 2.4. Treatment Progress

First, we referred the patient to the oral and maxillofacial surgery department for evaluation of the hypertrophic region of alveolar bone in the anterior mandible. No evidence of a pathologic condition was found, and it was considered to be a temporary phenomenon.

Based on this, we decided to start an orthopedic treatment and closely follow up the hypertrophic region. We placed a bonded rapid palatal expansion appliance with a hook for the face mask, and the screw was turned once a day for 2 weeks. The separation of the midpalatal suture was confirmed, and the face mask was set up with the instruction of wearing it for at least 14 h a day. After 5 months, the crossbite was relieved, and the anteroposterior jaw relationship was improved. Nevertheless, there was still a lack of space for the eruption of permanent teeth on the maxillary dentition (Figure 4A). We stopped the use of the face mask and set up an active removable appliance on the maxilla to gain space. After 18 months, sufficient space was made on the maxillary dentition, and we decided to finish the phase 1 treatment (Figure 4B and Figure 5).

However, the hypertrophic region of the alveolar bone in the anterior mandible did not disappear. We referred the patient to the oral and maxillofacial surgery department for the re-evaluation of the hypertrophic region and for the extraction of the impacted supernumerary tooth in the right mandible. A cone-beam computed tomography scan was conducted, and a biopsy of the hypertrophic region was performed. The radiologic and histopathologic findings revealed that the lesion consisted of mature cancellous bone and had no pathologic features (Figure 6). After extracting the supernumerary tooth, we decided to closely follow up the patient for observing the growth.

When the patient reached 13 years of age, he claimed that his teeth were still not fitting well. He showed a mild chin point deviation toward the right side, and severe lip canting while smiling was still present. The lateral facial profile was maintained favorably after the phase 1 treatment, but the Becker’s nevus area on the right cheek through the upper neck was more prominent, with hypertrichosis (Figure 7 and Figure 8). Intraorally, all permanent teeth except the second molars had successfully erupted; however, the hypertrophic region in the anterior mandible was still present. Therefore, he showed severe canting of mandibular anterior teeth with a 1 mm deviation of the apical base midline of the mandibular dentition toward the right side.

At this point, we noticed that the nevus, facial asymmetry, lip canting, and hypertrophy of alveolar bone occurred in the same area. Taking into consideration these symptoms, we suggested the presence of “Becker’s nevus syndrome” for the simultaneous occurrence of Becker’s nevus and other systemic anomalies. We interviewed the patient, and he informed us that he had been recently diagnosed with scoliosis by an orthopedist (Figure 8D). Fortunately, the severity of the scoliosis was incipient to moderate, with no need for active treatment but only a need for observation.

The patient was at stage 4 of the cervical vertebral maturation index in the deceleration phase of growth, and we decided to start phase 2 treatment. First, we decided to correct the canting of the mandibular anterior teeth by the intrusion of the right side with the segmental technique to avoid side effects. Due to the deep bite, it was not easy to place fixed braces on the mandibular anterior teeth. Therefore, we placed a fixed clear attachment with multiple clear buttons to splint the six anterior teeth of the mandible and started to intrusively rotate the anterior teeth with a mini screw, which was inserted between the mandibular right canine and the first premolar (Figure 9A).

After 4 months, the canting of the mandibular anterior teeth was much improved, and we placed fixed braces for comprehensive orthodontic treatment. We maintained the mild intrusive force on the mandibular right canine with an elastic chain during the leveling and alignment procedures to prevent relapse (Figure 9B), following which dental midline correction was performed with mini screws inserted between the first molar and second premolar of the right maxilla and left mandible (Figure 9C).

The appliances were removed after 19 months of phase 2 treatment. Fixed retainers were bonded to the lingual surfaces of anterior teeth in both arches. The maxillary and mandibular circumferential retainers were delivered with the anterior bite plate added to the maxillary retainers to prevent the relapse of the deep bite. We instructed the patient to wear them for 24 h a day for the next 6 months.

## 3. Results

Post-treatment photographs showed that the facial profile was improved, and ideal alignment and occlusion were achieved with a proper overjet and overbite (Figure 10, Figure 11 and Figure 12). The mandibular dental midline coincided with the maxillary dental midline and the facial midline. The canting of the mandibular anterior teeth was corrected with the intrusion of the mandibular right anterior teeth without a loss of periodontal support or root resorption. Although both the nevus and lip canting remained, the darkness of the hyperpigmentation was lightened by periodic laser treatment at a dermatology clinic.

Cephalometric analysis showed that a skeletal Class I relationship was well-maintained after the phase 1 treatment, with an ANB of 2.5°. Both previously lingually inclined maxillary and mandibular incisors were improved to the normal range. The patient was satisfied with the results and remained stable for 1 year after debonding. While a slight relapse pattern of open bite was observed in the left anterior region, no remarkable increase in the right mandibular right hypertrophic region was observed (Figure 13 and Figure 14).

## 4. Discussion

Becker’s nevus syndrome is clinically diagnosed by the presence of a Becker’s nevus along with other cutaneous, muscular, or skeletal abnormalities, including ipsilateral hypoplasia of the breast, supernumerary nipples, muscular dystrophy, scoliosis, or fused or accessory cervical ribs [20]. The exact etiology of Becker’s nevus syndrome is unclear, and the majority of the cases occur sporadically, with a low incidence of familial involvement [21]. This syndrome has been thought to be a hormone-dependent disorder because the overexpression of androgen receptors in the Becker’s nevus area has been detected [8]. Since androgens also affect hair, muscle, and bone development, this syndrome may be related to the pathogenesis of other systemic manifestations [14].

Several cases of anomalies in the maxillofacial region in young patients along with a Becker’s nevus consisting of facial asymmetry, unilateral bony or gingival hypertrophy on the maxilla, and tooth abnormalities have been reported [16,17,18]. In most cases, the patients did not show a progressive increase in the size or malignant transformation of the hypertrophic area. Our patient had a Becker’s nevus on the right cheek, chin, and upper neck area, along with scoliosis, facial asymmetry, and hypertrophy of the mandibular right anterior alveolar bone, which was different from the majority of the previous studies wherein the enlarged areas were reported on the maxilla. Due to the hypertrophy of the anterior alveolar bone of the mandible, our patient showed an anterior crossbite, which seemed to have induced growth inhibition in the premaxilla and led to a skeletal Class III relationship. Considering the patient’s young age and the cause of the skeletal Class III relationship, which was mainly an external factor rather than an intrinsic skeletal factor, the prognosis with orthopedic treatment was expected to be favorable [22]. The treatment result using a face mask was acceptable, and the patient showed no tendency for relapse throughout the treatment procedure.

At the first visit, the patient had no noticeable facial asymmetry; however, just before the beginning of phase 2 treatment, he showed a mild facial asymmetry with a chin point deviation toward the right side and continued to show severe lip canting while smiling. We examined the facial expression muscles of the patient, and found that he had a lack of function on lowering the right mouth corner, for which the depressor anguli oris and platysma are responsible [23]. The location of these muscles coincided with the Becker’s nevus area on the lower right face and upper right neck, and it was assumed that the facial asymmetry might be because of the asymmetry in muscle activity between the right and left sides of the face during the growth period.

For patients similar to the patient in this case, one of the most important considerations is to determine whether the hypertrophic area has a progressive feature, such as fibrous dysplasia, prior to the start of the orthodontic treatment. The biopsies of other patients with maxillary alveolar bone hypertrophy have revealed non-specific, non-inflammatory tissues, which were different from the findings in other pathologic diseases, such as fibrous dysplasia [24]. In particular, differentiating diagnosis from fibrous dysplasia is important because in the case of McCune–Albright syndrome, café-au-lait spots can be present along with fibrous dysplasia [25]. In our patient, the radiographic image of the involved area revealed mature cancellous bone in contrast to the “ground-glass” patterns of fibrous dysplasia [26]. In addition, consistent with the previous reports showing non-specific, non-inflammatory connective tissue hyperplasia, no pathologic features were observed in histopathology [24], and the size of the lesion was stable during the observation period. Therefore, based on these results, we decided to start phase 2 treatment, and it was possible to finish it with reasonable outcomes.

During the intrusion of the mandibular anterior teeth into the hypertrophic alveolar bone area in the anterior right mandible, we carefully monitored the patient to avoid side effects. Fortunately, he showed neither a loss of periodontal attachment nor root resorption of the mandibular anterior teeth after intrusion. Surprisingly, the size of the hypertrophic area diminished after orthodontic treatment and remained stable until the latest follow-up at 1 year after debonding. However, we observed an open bite at the left mandibular anterior area 1 year after debonding. During the segmental intrusion of the mandibular anterior teeth, intrusive force was applied to the right segment, and consequently, extrusion occurred at the left segment, which might have contributed to the relapse of the left anterior open bite. If the open bite tendency had been due to a progressive increase in the hypertrophic area, there would have been an open bite tendency in the posterior area as well. In addition, since a posteroanterior cephalogram was not taken after the phase 1 treatment in this case, the degree of change in the facial asymmetry of the patient during the phase 1 and phase 2 treatments could not be quantitatively confirmed through superimposition. Since orthodontic treatments in cases such as this are few, periodic follow-up and long-term monitoring are required in this case.

## 5. Conclusions

The present case shows successful results of orthopedic and orthodontic treatment in a young boy who had exhibited multiple characteristics of Becker’s nevus syndrome, including a Becker’s nevus, scoliosis, facial asymmetry, and alveolar bone hypertrophy in the anterior right mandible. We recommend that, when treating a complicated case such as this syndromic patient, systemic consideration and comprehensive approaches are necessary for predictable treatment results.

## Figures and Tables

**Figure 1 children-08-00072-f001:**
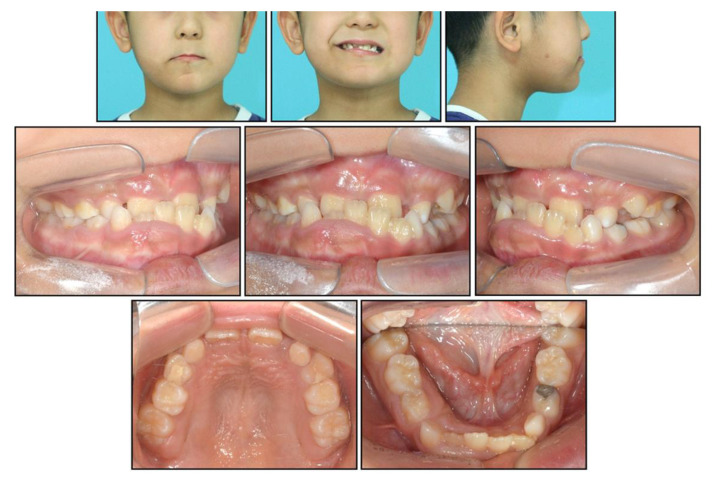
Pre-treatment facial and intraoral photographs.

**Figure 2 children-08-00072-f002:**
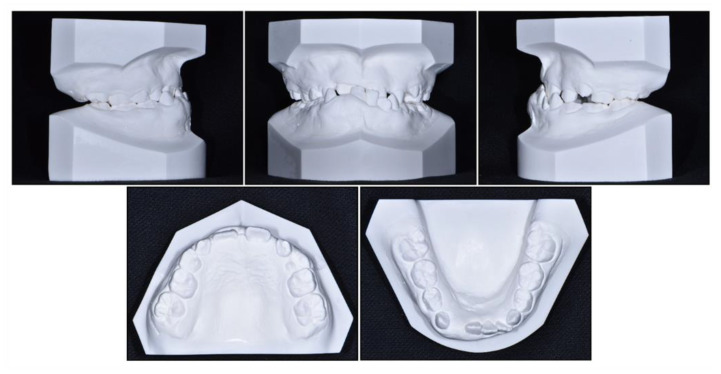
Pre-treatment cast models.

**Figure 3 children-08-00072-f003:**
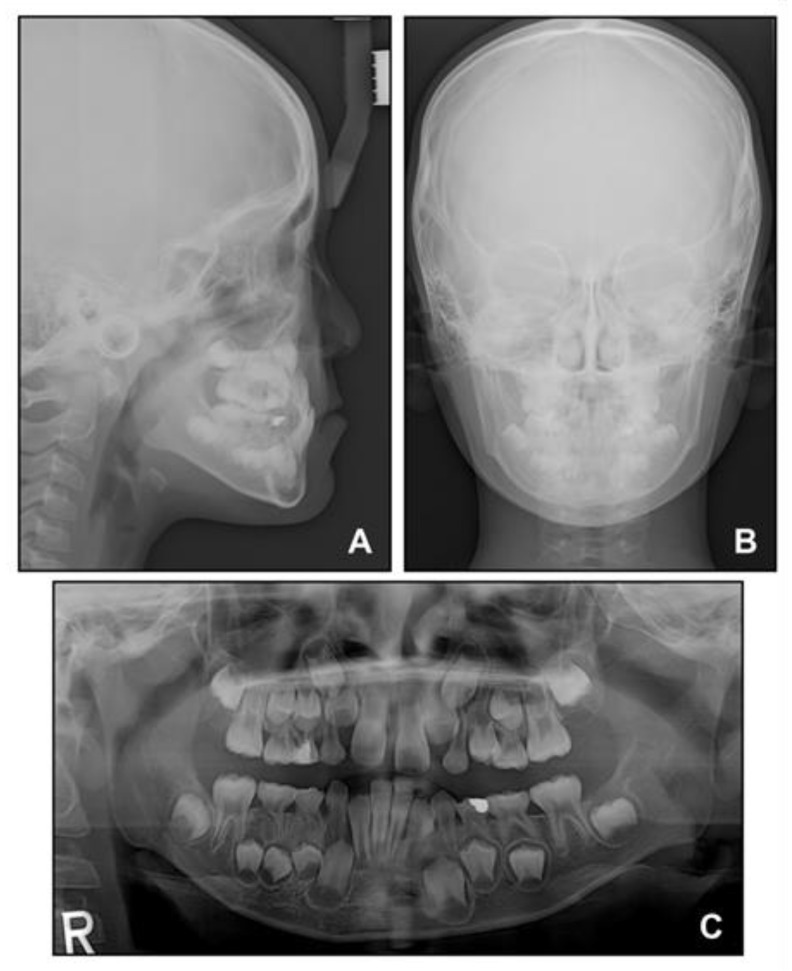
Pre-treatment radiographs: (**A**) Lateral cephalogram; (**B**) Posteroanterior cephalogram; (**C**) Panoramic radiograph.

**Figure 4 children-08-00072-f004:**
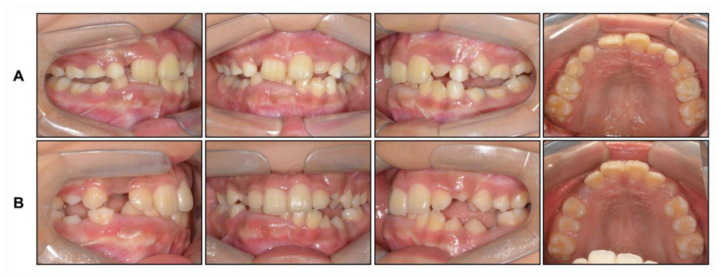
Intraoral photographs: (**A**) After face mask treatment; (**B**) At the end of phase 1 treatment.

**Figure 5 children-08-00072-f005:**
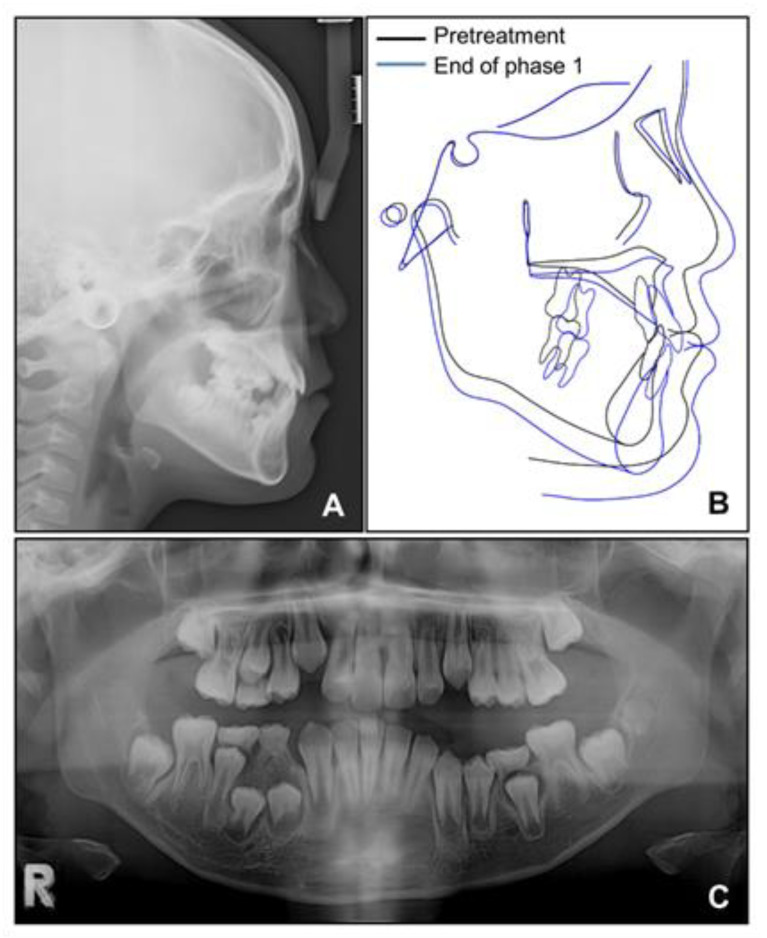
Radiographs after phase 1 treatment. (**A**) Lateral cephalogram; (**B**) Superimposition before and after phase 1 treatment; (**C**) Panoramic radiograph.

**Figure 6 children-08-00072-f006:**
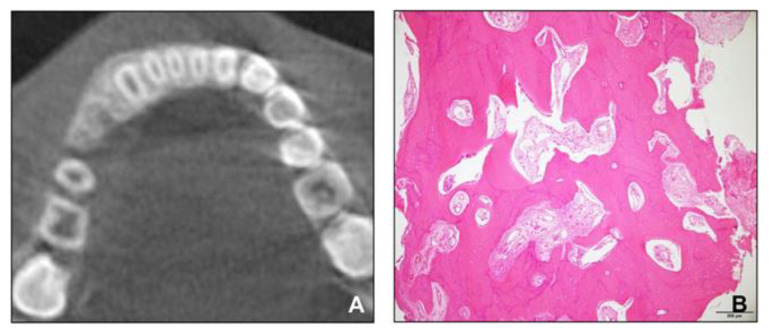
(**A**) Cone-beam computed tomographic image; (**B**), Histopathologic image showing mature cancellous bone.

**Figure 7 children-08-00072-f007:**
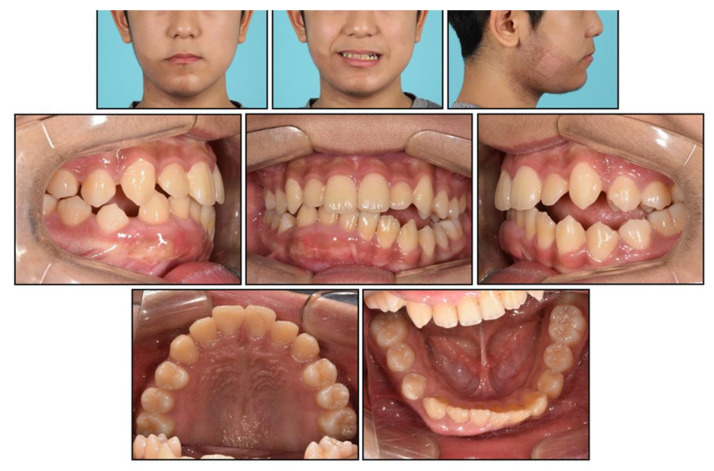
Facial and intraoral photographs before phase 2 treatment.

**Figure 8 children-08-00072-f008:**
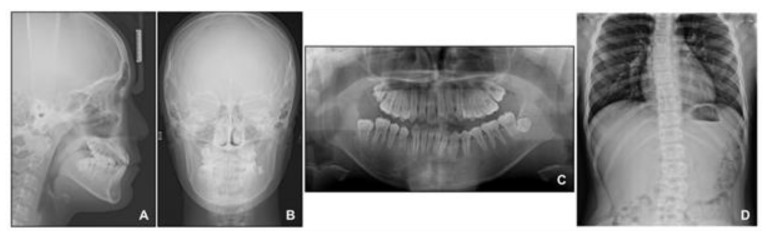
Radiographs before phase 2 treatment: (**A**) Lateral cephalogram; (**B**) Posteroanterior cephalogram; (**C**) Panoramic radiograph; (**D**) Standing anteroposterior radiograph of spine showing scoliosis.

**Figure 9 children-08-00072-f009:**
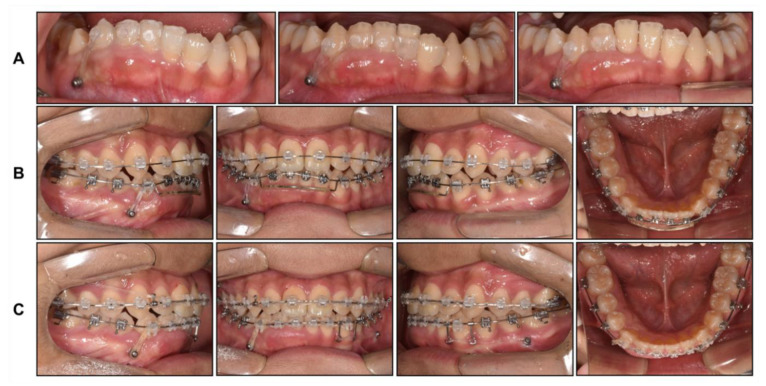
Intraoral photographs during phase 2 treatment: (**A**) Progressive intrusion of mandibular right anterior teeth using fixed clear attachment; (**B**) Leveling and alignment; (**C**) Midline correction using mini screws.

**Figure 10 children-08-00072-f010:**
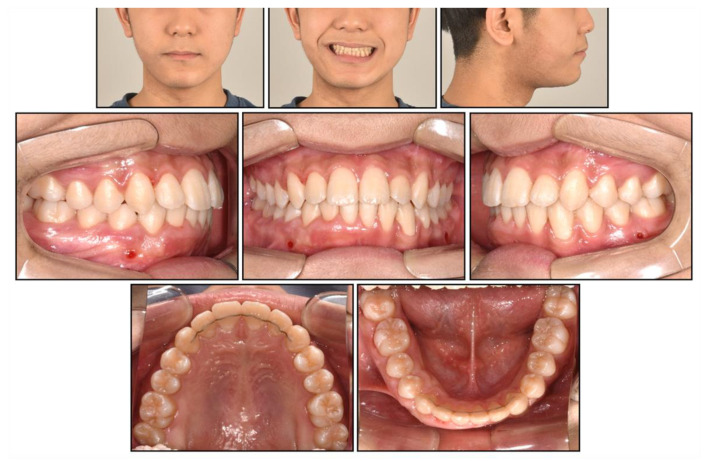
Post-treatment facial and intraoral photographs.

**Figure 11 children-08-00072-f011:**
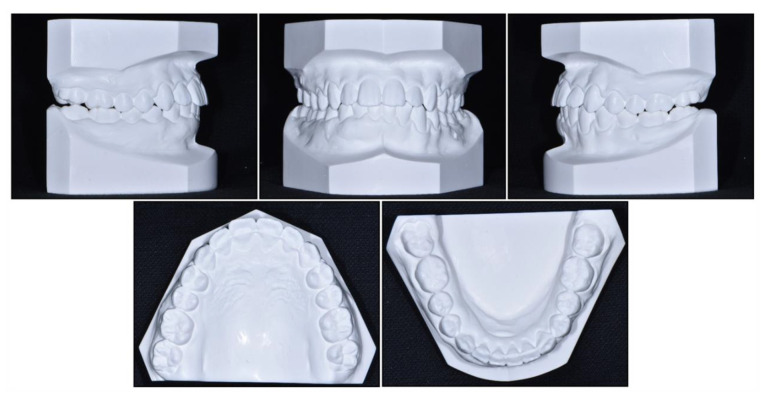
Post-treatment cast models.

**Figure 12 children-08-00072-f012:**
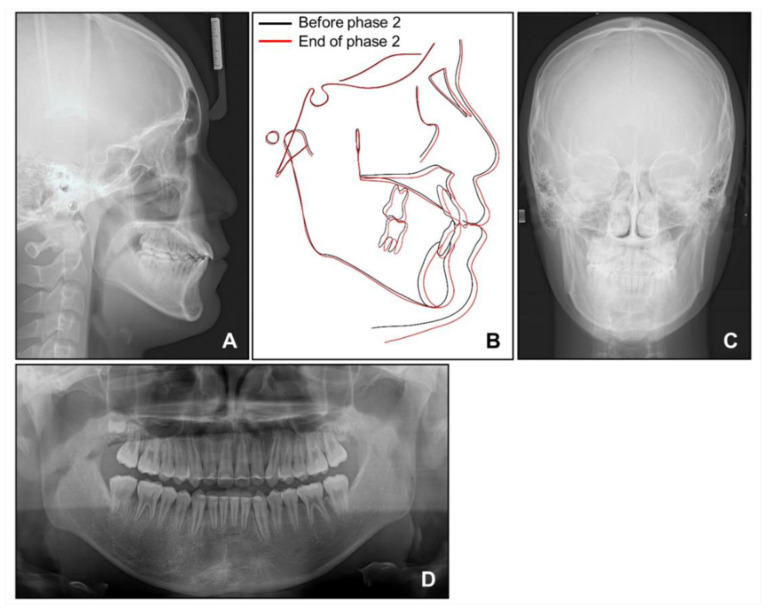
Post-treatment radiographs: (**A**) Lateral cephalogram; (**B**) Superimposition before and after phase 2 treatment; (**C**) Posteroanterior cephalogram; (**D**) Panoramic radiograph.

**Figure 13 children-08-00072-f013:**
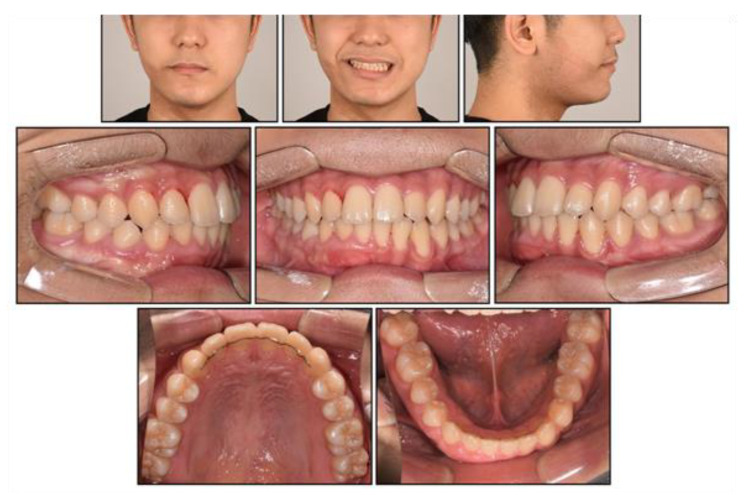
Facial and intraoral photographs after 1-year retention.

**Figure 14 children-08-00072-f014:**
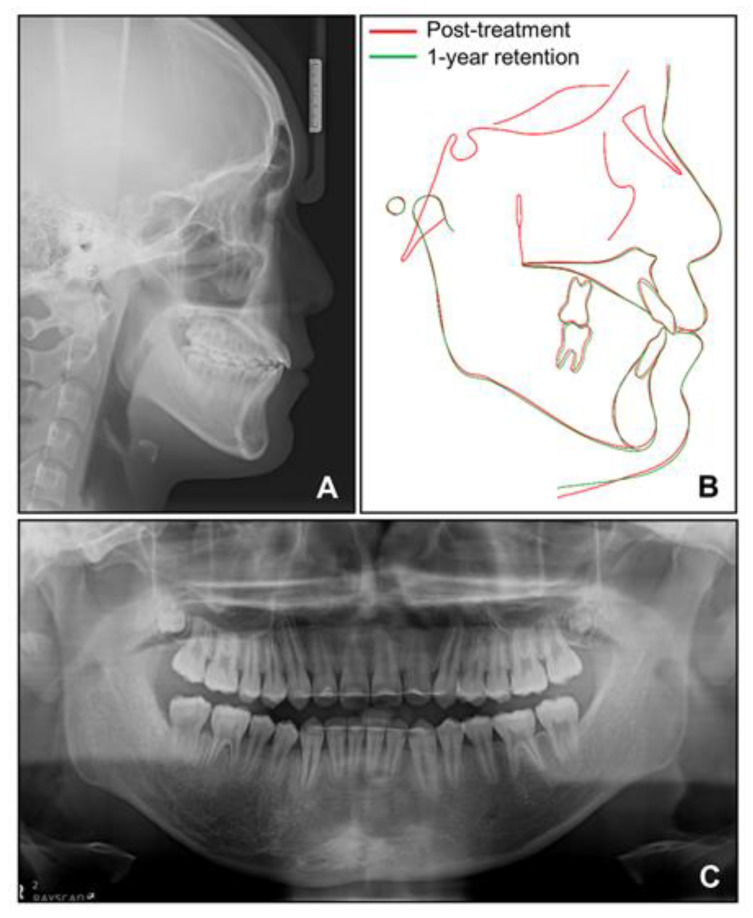
Radiographs for 1-year retention. (**A**) Lateral cephalogram; (**B**) Superimposition between post-treatment and 1-year retention; (**C**) Panoramic radiograph.

**Table 1 children-08-00072-t001:** Lateral cephalometric measurements.

Measurement	Normal Value	Pre-Treatment (8 y 2 mo)	After Phase I Treatment (11 y)	Before Phase II Treatment (12 y 9 mo)	Post-Treatment (14 y 5 mo)	1-Year Retention (15 y 5 mo)
Skeletal						
SNA (°)	81.0 ± 3.0	75.4	77.3	78.8	80.5	80.6
SNB (°)	78.0 ± 3.0	75.7	75.4	76.2	78.0	78.2
ANB (°)	4.0 ± 2.0	−0.3	1.9	2.6	2.5	2.4
Wits (mm)	−2.0 ± 2.4	−4.5	−2.6	−0.3	−0.9	−0.9
SN-GoMe (°)	36.0 ± 4.0	37.8	38.2	38.0	36.9	37.0
Gonial angle (°)	122.0 ± 6.0	127.2	127.8	128.1	128.3	127.9
Dental						
U1 to SN (°)	105.0 ± 5.0	86.1	101.5	105.4	109.8	110.0
L1 to GoMe (°)	95.0 ± 4.0	87.0	84.3	87.4	96.3	95.3
Soft tissue						
Nasolabial angle (°)	94.4 ± 10.3	105.3	102.9	97.3	95.8	96.9
Upper lip to E line (mm)	2.0 ± 2.0	−2.7	0.3	−0.4	−0.6	−0.9
Lower lip to E line (mm)	4.0 ± 3.0	1.0	1.1	1.0	0.0	−0.5

SNA, angle of the lines connecting the sella, nasion, and point A; SNB, angle of the lines connecting the sella, nasion, and point B; ANB, angle of the lines connecting point A, the nasion, and point B; SN, the plane formed by the sella and nasion; GoMe, the plane formed by the gonion and menton; U1, upper central incisor; L1, lower central incisor; E line, a line drawn from the pronasale to soft tissue pogonion.

## Data Availability

All relevant data are within the manuscript.

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
