# Peer review of "Comprehensive Treatment of a Skeletal Class III Growing Patient with Alveolar Bone Hypertrophy in the Anterior Mandible Associated with Becker’s Nevus Syndrome: A Case Report"

_children, 2021, doi:10.3390/children8020072_

Round 1
Reviewer 1 Report
In the first place, I would like to congratulate the authors for the effort put into the article.
Even though the Becker nevus syndrome is not a very frequent syndrome in children, I believe that this case report and the treatment has been carried out in an excellent way and it can be very useful as a guide for other clinicians in the treatment of patients with similar pathology.
English style and grammar should be revised for minor changes
Author Response
We thank the reviewer for positive comments on our manuscript. The manuscript has been checked and edited for English language by a professional English language editing company.
Reviewer 2 Report
I have no comment. The clinical case is interesting and well presented.
Author Response
We thank the reviewer for a positive feedback on our manuscript.
Reviewer 3 Report
Thank you very much for the opportunity to review this very well done manuscript. I find the content to be very interesting and the quality of the writing very well done.
Author Response

(The authors gave the same response as above.)
